# Discovering Potential in Non-Cancer Medications: A Promising Breakthrough for Multiple Myeloma Patients

**DOI:** 10.3390/cancers16132381

**Published:** 2024-06-28

**Authors:** Omar S. Al-Odat, Emily Nelson, Tulin Budak-Alpdogan, Subash C. Jonnalagadda, Dhimant Desai, Manoj K. Pandey

**Affiliations:** 1Department of Biomedical Sciences, Cooper Medical School of Rowan University, Camden, NJ 08103, USA; omaral98@students.rowan.edu (O.S.A.-O.); nelson52@students.rowan.edu (E.N.); 2Department of Chemistry and Biochemistry, Rowan University, Glassboro, NJ 08028, USA; jonnalagadda@rowan.edu; 3Department of Hematology, Cooper Health University, Camden, NJ 08103, USA; budak-alpdogan-tulin@cooperhealth.edu; 4Department of Pharmacology, Penn State Neuroscience Institute, Penn State College of Medicine, Hershey, PA 17033, USA; ddesai@pennstatehealth.psu.edu

**Keywords:** hematological malignancies, multiple myeloma, drug repurposing, drug development, drug resistance

## Abstract

**Simple Summary:**

Multiple myeloma (MM) is a type of cancer that affects the blood and bone marrow. Each individual diagnosed with MM will inevitably experience a relapse or develop resistance to the prescribed treatment. The development of new pharmaceuticals is an expensive and time-consuming process, which requires the investigation of more efficient approaches. This review article explores the potential of repurposing current drugs, originally designed for other conditions, for the treatment of MM. This approach is more efficient and economical in comparison to the process of developing new drugs from scratch. For instance, thalidomide, initially used for several medical ailments, has shown effectiveness in treating MM. This study emphasizes the potential of repurposing common drugs, such as aspirin and statins, for the treatment of MM. This approach not only speeds up the availability of new treatments but also offers hope for better outcomes for patients with MM. Future investigations will give priority to determining the most effective dosages and integrating these repurposed drugs with traditional therapy to improve their effectiveness.

**Abstract:**

MM is a common type of cancer that unfortunately leads to a significant number of deaths each year. The majority of the reported MM cases are detected in the advanced stages, posing significant challenges for treatment. Additionally, all MM patients eventually develop resistance or experience relapse; therefore, advances in treatment are needed. However, developing new anti-cancer drugs, especially for MM, requires significant financial investment and a lengthy development process. The study of drug repurposing involves exploring the potential of existing drugs for new therapeutic uses. This can significantly reduce both time and costs, which are typically a major concern for MM patients. The utilization of pre-existing non-cancer drugs for various myeloma treatments presents a highly efficient and cost-effective strategy, considering their prior preclinical and clinical development. The drugs have shown promising potential in targeting key pathways associated with MM progression and resistance. Thalidomide exemplifies the success that can be achieved through this strategy. This review delves into the current trends, the challenges faced by conventional therapies for MM, and the importance of repurposing drugs for MM. This review highlights a noncomprehensive list of conventional therapies that have potentially significant anti-myeloma properties and anti-neoplastic effects. Additionally, we offer valuable insights into the resources that can help streamline and accelerate drug repurposing efforts in the field of MM.

## 1. The Significance of Repurposing

Every year, millions of people around the world are diagnosed with cancer, and sadly, many lose their lives to this devastating disease. Globally in 2020, hematological malignancy incidence was almost 1.3 million and mortality was over 700,000, making hematological cancers the fourth highest in terms of cancer mortality [1]. Multiple myeloma (MM) accounts for more than 12% of all hematologic cancers. MM is a complex form of cancer that can be challenging to diagnose and treat. It is a malignancy of plasma B cells and originates in the bone marrow (BM). Despite advancements in treatments, MM remains an incurable disease as it will inevitably progress or develop resistance to treatments in all patients. Optimistically speaking, however, therapeutic advancements could potentially increase life expectancy for MM patients, addressing both current and future challenges.

Drug research and development, especially in the realm of cancer drugs, have experienced remarkable changes in recent decades. Year after year, the collection of medications to combat cancer continues to grow. Nevertheless, there are a number of challenges that need to be addressed before a drug can be brought to market. These include the extensive drug development process, the significant expenses associated with drug research, the possibility of adverse events, and the limited efficacy of new treatments. In the field of cancer treatment, the search for effective lead compounds has long followed a well-established process. This involves conducting extensive preclinical and clinical research to carefully assess and document the compounds’ pharmacological properties, anti-cancer effects, and potential toxicity (Figure 1).

Despite the advancements in technology and our improved understanding of human disease, the translation of these advantages into therapeutic breakthroughs has been disappointingly slow [2,3]. The global pharmaceutical industry faces a multitude of challenges, such as high attrition rates [4,5], evolving regulations, and prolonged time to market for new drugs in certain therapeutic fields, all of which have led to increased costs. A prediction has been made that the return on investment for new drug development is expected to be less than one dollar for every dollar spent due to the increasing cost and time required for these endeavors [6]. This could potentially reduce the attractiveness of the pharmaceutical industry as an investment option. As shown in Figure 1, the process of developing a new medicine can span a decade or more and require a significant investment of billions of dollars [7,8]. Thus, it is crucial to explore alternative methods for drug development.

Drug repurposing, also referred to as drug repositioning, re-tasking, or reprofiling, is a strategy employed to uncover new therapeutic uses for medications that have already been approved, tested, or are currently available on the market. These medications were initially developed and trialed for different therapeutic purposes than the ones they are being explored for now. Compared to new drug development, the repurposing approach offers numerous advantages to the research and clinical fields. It can be described as finding alternative applications for familiar medications. Previous clinical trials have confirmed the safety, effectiveness, and efficiency of a drug, considering its pharmacokinetic, pharmacodynamic, and toxicological properties. As a result, it has gained popularity and become more accessible. Looking at a drug molecule from a fresh angle could help us uncover new possibilities for its therapeutic applications beyond the conventional ones. Repurposing a medicine that has already received approval for a different therapeutic indication requires less funding. The process of approving drugs through the repurposing approach is estimated to take anywhere from 3 to 12 years and cost between USD 40 million to USD 80 million, which is significantly lower than the cost of the traditional drug development method [9,10]. In addition, it is anticipated that the repurposing pathway will have a higher approval rate for medications, estimated at around 30%, compared to the typical drug development method which only yields 10% [10]. Once failures are taken into consideration, these advantages can lead to a faster and safer return on investment for the development of repurposed pharmaceuticals, along with lower average development costs. Only a small fraction, around 5%, of oncology treatments that qualify for phase 1 clinical trials end up getting the green light from the FDA. The chances of potential anti-cancer therapies being approved are even more dismal, with only 1 in 5000–10,000 making the cut [11]. Through the implementation of a drug repurposing strategy, it becomes feasible to bypass the initial phase and make swift progress to subsequent phases of clinical trials, resulting in a reduction in concerns related to pharmacodynamics [11]. Therefore, there is a strong attraction towards methods that optimize the utilization of information from drugs that have already been approved and brought to market [12]. In the end, repurposed pharmaceuticals have the potential to uncover new targets and pathways for further exploration. Therefore, researchers are currently focusing on exploring new pharmacological action mechanisms that have emerged from unexpected clinical trial findings. These findings have sparked interest in bridging the gap between clinical practice and laboratory research. Various preclinical investigations are conducted to validate the claim of a new pharmacological indication. This approach focuses on addressing chronic illnesses such as diabetes, cancer, and other rare disorders [12]. There are numerous benefits to repurposing pharmaceuticals in general. This approach is both compelling and practical, especially in today’s era in which deep data mining technologies are readily accessible. The frequent approval of repurposed pharmaceuticals suggests that the technique of repurposing has a minimal risk of drug failure, as most of the drugs being repurposed undergo thorough safety testing. The drug repurposing sectors have experienced consistent growth from a business perspective, with approximately 14–16 new companies emerging every five years [13,14].

MM is an incurable cancer in which relapse inevitably occurs even for patients in remission. Typically, the primary treatment for newly diagnosed MM includes bortezomib (a proteasome inhibitor), lenalidomide (an immunomodulatory agent), and dexamethasone (a corticosteroid) [15]. This combination of drugs is termed the VRd regimen and has shown to be effective in the treatment of MM upfront [15]. Patients may also undergo autologous stem cell transplantation, if deemed eligible, either initially or later within the treatment process [15]. However, this regime is not permanently curative, and patients must switch drugs when they develop resistance. Malignant plasma cells display high levels of aberrant cell signaling pathways that prevent apoptosis and promote cell survival, ultimately leading to relapse. The NF-κB pathway is activated by cytokines, such as IL-6, and the binding of MM cells to bone marrow stromal cells [16]. This pathway is inhibited by proteasome inhibitors and strongly linked with disease progression as it increases MM cell proliferation and resists apoptosis [16]. Similarly, the antiapoptotic subgroup of Bcl-2 includes Mcl-1, Bcl-2, and Bcl-xL, which are commonly overexpressed in MM and are shown to prevent intrinsic apoptosis by suppressing BH-3 activators and competing for binding Bax and Bak proteins [17]. The interaction of myeloma cells with the bone marrow microenvironment is integral to malignant myeloma cell proliferation and the development of resistance [16]. Cytokines such as IL-6, IL-3, and IL-5 and growth factors such as VEGF and EGF trigger upregulation of intracellular pathways such as NF-κB, JAK/STAT, PI3-K/Akt, and Bcl-2 antiapoptotic proteins—all of which promote MM cell proliferation and therefore the development of resistance [17]. Figure 2 illustrates the primary signaling pathways that play a crucial role in the development of MM.

MM therapy has stood out as an anomaly in terms of the typical balance between risks and benefits. Some medications used in anti-cancer chemotherapy and radiation have severe adverse effects that can be life-threatening. The primary objective of therapy in MM is to ensure the patient’s survival. Overcoming resistance to treatment is a significant hurdle when it comes to developing a successful dosage plan. This review explores the potential of various medications to treat MM, highlighting the anti-cancer properties of conventional drugs such as thalidomide, statins, celecoxib, aspirin, artesunate, leflunomide, rapamycin, nelfinavir, valproic acid, metformin, bisphosphonates, and clarithromycin. We have compiled a comprehensive list of resources that may prove valuable for drug repurposing.

## 2. Pharmacological Repurposing Strategies and Tools

The drug repurposing strategy involves three stages that occur before a candidate drug progresses through the development pipeline. These stages include generating hypotheses to identify a potential molecule for a specific indication, assessing the drug’s effects using preclinical models, and conducting phase II clinical trials to evaluate its efficacy (assuming sufficient safety data have been gathered from phase I trials conducted for the original indication). The first stage is crucial. There are different types of systematic approaches, such as computational approaches and experimental approaches, that are being increasingly used together (Figure 3). These two core domains encompass clinical data-driven drug repurposing. These techniques generate hypotheses for repurposing by thoroughly analyzing various types of data, such as electronic health records (EHRs), genotyping, chemical structure, gene expression, or proteomic information [18]. The process of signature matching involves comparing the unique characteristics or “signature” of a specific pharmaceutical compound with those of another compound, disorder, or clinical phenotype [19,20]. The creation of a drug’s signature can be derived from three different sources of data: transcriptome (RNA), proteomic, or metabolomic data; chemical structures; or adverse event patterns.

Methods rooted in systems biology, like the Genome-wide Positioning Systems network (GPSnet), are aiding in the comprehension of disease–gene–drug interactions, potentially facilitating drug repurposing for MM [13,21,22,23,24,25]. In addition, there are multinational collaborative programs, like Repurposing Drugs in Oncology (ReDO), that are working to accelerate the repurposing of non-cancer drugs for cancer treatment. Table 1 provides supplementary information and tools that can assist in the drug repurposing process. Through the utilization of biological databases and systems biology technologies, it becomes feasible to identify driver pathways in MM. Subsequently, pre-existing drugs from drug libraries can be discovered, which possess the ability to target these driver pathways [26]. For example, by utilizing resources like as STRING and KEGG, the identification of celecoxib as a viable therapy for MM was achieved. This was accomplished by specifically targeting the COX-2 enzyme in MM cells. DeSigN had a role in identifying thalidomide by uncovering its impact on the IKK and NF-κB pathways in MM. Leflunomide was discovered using the Cancer Cell Line Encyclopedia because it inhibits DHODH. CMap analysis revealed that metformin activates AMPK and regulates cell cycle proteins, indicating its potential for treating MM.

## 3. Medicines That Could Be Repurposed to Treat MM

MM cells exhibit a wide range of abnormal signaling pathways and protein expression that contribute to the progression of the disease. Fortunately, there are several agents that show promise in treating MM by targeting these pathways, either directly or indirectly. Several drugs being considered for repurposing have mechanisms of action that may have potential in treating various types of cancers. However, these agents are especially intriguing when it comes to MM. These findings indicate that MM cells exhibit abnormal signaling pathways, chemokines, cytokines, and proteins that are targeted by these agents, many of which play a critical role in disease relapse. Given the inevitability of relapse in MM patients, it is crucial to explore alternative or additional medications. One promising approach is drug repurposing, which offers an efficient and potentially effective way to address this challenge. Table 2 displays the medications that have potential to demonstrate effectiveness against MM. Drug repurposing has often occurred by chance and circumstance throughout history. The identification of an off-target impact or a newly identified on-target effect led to the advancement of a pharmaceutical drug towards commercialization. The logic behind estimating the anti-cancer properties of non-cancer drugs was established based on their effectiveness, mechanisms of action, and minimal side effects (Figure 4). Interestingly, the most successful cases of drug repurposing so far have not followed a systematic approach. As an example, the discovery of thalidomide’s effectiveness in treating erythema nodosum leprosum (ENL) and MM was purely accidental [2].

### 3.1. Thalidomide

Thalidomide, a derivative of glutamic acid [86], was initially introduced as a sedative in 1957 [87] and also employed as an antiemetic during pregnancy. However, due to its severe teratogenic effects, it was withdrawn from the market in 1961 [88]. Moving forward, however, thalidomide’s efficacy against erythema nodosum leprosum (ENL), a cutaneous form of leprosy, led to its FDA approval for this condition in 1998 [89]. Subsequently, its anti-angiogenic properties expanded its usage to treat various skin disorders, infectious diseases, immunologic and rheumatologic disorders, hematologic diseases, and several cancer types [90,91,92]. In 2006, the FDA-approved thalidomide for newly diagnosed MM in combination with dexamethasone. Thalidomide influences numerous signaling pathways commonly dysregulated in cancer cells [93]. It inhibits tumor necrosis factor (TNF), frequently deregulated in cancers [94], and impedes NF-κB activation, a protein complex crucial for DNA transcription, cytokine production, and cell survival, by inhibiting IκB kinase (IKK) [95]. Additionally, thalidomide inhibits interleukin (IL)-1, IL-6, and IL-12, granulocyte-macrophage colony-stimulating factor (GM-CSF), vascular endothelial growth factor (VEGF), basic fibroblast growth factor (bFGF), and interferon (IFN) [96,97], making it potentially effective against various cancers like AIDS-related Kaposi’s sarcoma, renal cell carcinoma, and gliomas [98,99,100]. Thalidomide shows promise in treating hormone-dependent prostate cancer, as evidenced by a phase III clinical trial where it reduced time to prostate-specific antigen (PSA)-based progression in patients receiving androgen deprivation therapy [101]. However, toxicity issues have impeded many clinical trials of thalidomide as a cancer treatment [102,103,104], leading to the development of less toxic thalidomide analogues with similar anti-cancer effects. Thalidomide’s repurposing involved off-label usage, pharmacological analysis, and derivative development, resulting in significant clinical and commercial success in MM.

### 3.2. Statins

The discovery of statins as HMG-CoA reductase inhibitors led to their approval for treating individuals at high risk of heart failure. The transformation of HMG-CoA to mevalonic acid is catalyzed by HMG-CoA reductase, which is the first and rate-limiting step in cholesterol production. Statins inhibit HMG-CoA reductase, which in turn suppresses cholesterol biosynthesis and the mevalonate cascade. Byproducts of this cascade play a vital role in the survival of cancer cells; therefore, the effectiveness of statins in combating various types of cancer cells has been assessed. Studies have demonstrated that Simvastatin can induce apoptosis in CML cells [105]. In addition, it has been found through meta-analyses that statins are associated with a lower risk of hematological malignancies [106,107]. The use of statins has been linked to an enhanced survival rate and a decrease in mortality in MM [53,54,55,56]. Clinically in 4315 patients, statin use is associated with improved survival in MM [54]. A PRISMA-compliant meta-analysis indicated that statin use might be a protective factor for MM incidence [108].

MM cells containing chromosomal translocation t(4;14) is typically a more aggressive subtype of MM. It has been shown that t(4;14)-positive cells have increased reliance on the mevalonate pathway due to its production of geranylgeranyl pyrophosphate [109]. When treated with statins, t(4;14)-positive cells underwent an integrated stress response due to a lack of geranylgeranyl pyrophosphate and, used in conjunction with bortezomib, results showed even greater cytotoxic effects in vivo [109]. MM cells frequently rely heavily on Mcl-1 for their survival, making them resistant to venetoclax, a Bcl-2 inhibitor [17]. Statins have the potential to overcome resistance to venetoclax in MM cell lines and primary cells by blocking the mevalonate pathway. Furthermore, it has been observed that statins can enhance the susceptibility to apoptosis induced by S63845, a potent Mcl-1 inhibitor [110]. In retrospective analyses of venetoclax clinical trials in patients with MM, the use of statins prior to treatment was found to be significantly associated with a greater likelihood of achieving a rigorous complete response prolonging progression-free survival [110]. The sensitivity of MM cells to venetoclax is increased by statins through the upregulation of two pro-apoptotic proteins, PUMA (in a mechanism not dependent on p53) and NOXA (via the integrated stress response) [110]. Additional research involving animals and humans suggests that statins may offer some level of protection against the development of MM [56]. This study presents a new avenue for utilizing statins in the treatment of MM and other blood malignancies. It also indicates their potential in preventing cancer in susceptible populations.

### 3.3. Celecoxib

Several studies have demonstrated a clear link between the overexpression of cyclooxygenase-2 (COX-2) and lower survival rates in patients with MM [111,112]. Hematological malignancies such as Hodgkin’s lymphoma, NHL, CLL, CML, and MM show an upregulation of COX-2. COX-2 plays a crucial role in angiogenesis, metastasis, cell proliferation, and survival [57,113]. Celecoxib, a non-steroidal anti-inflammatory drug (NSAID), selectively inhibits COX-2. Celecoxib has been used in the clinic since 1998 to treat osteoarthritis, rheumatoid arthritis (RA), and generalized pain. In addition, extensive research has indicated celecoxib as a promising chemopreventive drug for various forms of cancer [58]. Furthermore, studies have indicated that celecoxib has the ability to inhibit crucial survival proteins in various types of cancers, such as Mcl-1, Bcl-2, survivin, and Akt [114]. In MM, these proteins have all been established as overexpressed; additionally, Mcl-1 and Bcl-2 are drivers for the mechanisms of acquired resistance and relapse [17]. The FDA has previously approved celecoxib for use in a number of cancer therapies, including familial adenomatous polyposis (FAP) (800 mg/day), albeit at higher doses than prescribed for RA or osteoarthritis [115]. Higher dosages of celecoxib can lead to various side effects, including cardiotoxicity, gastrointestinal issues, and renal problems. Therefore, it is important to exercise caution when using NSAIDs in cancer treatment. Limited data exist regarding the role of COXs in the onset and progression of MM. To address this, researchers tested seven human MM cell lines representing different disease stages to assess COXs expression levels and cell viability upon treatment with COX inhibitors alone or in combination with conventional anti-MM drugs. The results showed that all tested drugs had moderate antiproliferative effects on MM cell lines [116]. Celecoxib shows great potential as a potential anti-myeloma drug, considering its correlation with decreased survival rates in MM due to COX-2 overexpression.

### 3.4. Aspirin

Aspirin, also known as acetylsalicylic acid (ASA), is another, more common, NSAID that is utilized for pain relief and reducing fever. In addition, it effectively helps prevent myocardial infarctions and cerebrovascular accidents caused by thromboembolism. As a target, aspirin is not as selective as celecoxib. Aspirin inhibits both COX-1 and COX-2, preventing the synthesis of prostaglandin H2 (PGH2) and prostaglandin E2 (PGE2). These compounds play a role in regulating inflammatory and thrombus formation processes. Aspirin not only inhibits COX-1/2, but also has a suppressive effect on other inflammatory cytokines like NF-κB. Additionally, it impacts pro-survival ERK signaling in cancer cells. It is worth noting that MM cells exhibited an overexpression of both NF-κB and ERK signaling [117,118]. Furthermore, Aspirin demonstrates its anti-tumor effect in MM by inhibiting Blimp1 and activating the ATF4/CHOP pathway [119]. Based on two significant prospective studies, the “Health Professional Follow-up Study” (1986–2008) and the “Nurses’ Health Study” (1976–2008), it has been found that taking aspirin after being diagnosed with MM is associated with a lower risk of death, both overall and specifically related to MM [60]. According to the trials, regularly taking 325 mg aspirin five or more times per week was linked to a significant 40% decrease in the rate of MM [61]. However, it is important to consider potential secondary issues like bleeding stomach ulcers and heartburn when using aspirin for an extended period of time.

### 3.5. Clarithromycin

Clarithromycin (CAM) is a widely used antibacterial treatment belonging to the class of semisynthetic macrolide antibiotics. The first clinical study revealed a significant increase in median survival time for patients with advanced non-small cell lung cancer (NSCLC) who underwent long-term CAM therapy [120]. It has been discovered that CAM can be a helpful addition to the treatment of MM [68]. Although treating MM with single agent CAM therapy is not effective, there is confirmed efficacy of CAM in combination chemotherapies for the treatment of MM. There have been several documented clinical trials that have explored the use of CAM in combination with other drugs for MM patients. These trials include NCT01745588, NCT01559935, and NCT02248428 [69,121,122,123,124,125,126,127]. However, most recently, a randomized phase III clinical trial found that in combination with steroid dexamethasone and lenalidomide, CAM did not improve progression-free survival due to the toxic effects of steroid overexposure (NCT02575144) [128].

It is widely recognized that various myeloma growth factors (MGFs), including IL-6, have a significant impact on the advancement of MM [17,129]. CAM has been shown to inhibit several MGFs, including IL-6 [130]. The treatment of MM with complementary and alternative medicine CAM involves various potential mechanisms of action. These include autophagy inhibition, immunomodulatory activity, reversibility of drug resistance, steroid sparing/enhancing impact, and suppression of MGFs. MM is characterized by the excessive growth of cancerous plasma cells that produce a single type of immunoglobulin (Ig). The presence of excessive misfolded or unfolded Ig can cause considerable stress on the endoplasmic reticulum [131]. Thus, MM arises as a fragile tumor that is particularly susceptible to autophagy inhibitors, proteasome inhibitors, and histone deacetylase 6 inhibitors. The combined effects of CAM play a crucial role in the treatment of MM [132].

### 3.6. Rapamycin

The antifungal agent rapamycin, also known as sirolimus and commercialized as Rapamune^®^, was initially discovered by Surendra Nath Sehgal and colleagues. It was isolated from the bacteria Streptomyces hygroscopicus on the island of Rapa Nui [133]. One of the initial signs of its approval was its use in managing graft rejection in kidney transplant recipients, thanks to its immunosuppressant properties [134]. Rapamycin was later identified as an antagonist of the mTOR (mammalian target of rapamycin) signaling pathway. mTOR plays a crucial role in various signaling pathways, such as cytoskeleton maintenance, protein synthesis, autophagy, lipid synthesis, cell growth, and angiogenesis [135]. In addition to its role in preventing graft rejection, rapamycin has been extensively researched as a potential anti-cancer treatment for various types of cancer. Rapamycin sensitized MM cells to apoptosis induced by dexamethasone [70]. In addition, the effectiveness of analogs and newer TORC1/TORC2 inhibitors in myeloma models has been supported by preclinical studies. Clinical trials in the initial phases have already begun [71]. Rapamycin, when combined with CC-50 (Revlimid), an immunomodulatory analog (IMiD) of thalidomide, has demonstrated anti-MM activity [72]. These trials serve as the basis for evaluating the effectiveness of mTOR inhibitors in combination with IMiDs in improving patient outcomes in MM. The individuals under investigation experienced negligible adverse effects from rapamycin, which highlights its decreased risk and treatment advantages. In order to further improve the efficiency of this molecule, efforts have been made to enhance its structure. Everolimus, an orally administered inhibitor of the mTOR pathway, is a second-generation compound derived from rapamycin (sirolimus). It has been investigated in the treatment of MM, although it does not demonstrate much effectiveness when used alone. The concurrent administration of lenalidomide and everolimus was well received by patients, as it resulted in expected side effects and demonstrated positive outcomes in a group of individuals who had undergone extensive prior treatment [73]. This research can provide guidance in selecting patients for future clinical studies involving mTOR inhibition in MM. 

### 3.7. Valproic Acid

Valproic acid (VPA, Depakene) is a medication that is commonly prescribed for the treatment of migraine, seizures, epilepsy, and bipolar disorders. Valproic acid is believed to block voltage-gated sodium channels and histone deacetylases (HDAC). HDAC plays a crucial role in the survival of MM cells. It is believed that the impact of bortezomib on apoptosis in MM is partially due to the inhibition of Class-I HDACs [74]. In addition, VPA has been found to inhibit the activation of NF- κB and the production of inflammatory cytokines TNF and IL-6 [136]. It has been found that VPA enhances autophagic flux in human MM cells [137]. In vitro studies demonstrated autophagy activation in MM cell lines RPMI8226 and U266 following VPA treatment [138]. There are indications that VPA could potentially be utilized as a treatment for MM [139].

### 3.8. Nelfinavir

Nelfinavir, an efficient protease inhibitor, is effective against both HIV-1 and HIV-2. As for ritonavir, principally attributed to the drug’s anti-cancer properties include its ability to disrupt Akt signaling and induce endoplasmic reticulum stress. Nelfinavir therapy has been seen to induce regression of prostate xenograft tumors and decrease phosphorylation of both STAT3 and Akt [140]. In MM, treatment induces a reduction in signaling via Akt, STAT3, and Erk1/2 by inhibiting the 26S proteasome [75]. Additionally, Nelfinavir and bortezomib have a synergistic effect. In a trial involving MM and non-small cell lung cancer, the combination of bortezomib and nelfinavir increased endothelial resistance (ER) stress and inhibited growth in vitro and in vivo. The buildup of ubiquitin (Ub) proteins is the mechanistic consequence of nelfinavir therapy. Bortezomib, on the other hand, inhibits degradation by the proteasome; this interferes with proteotoxic stress and ultimately induces cell death [141]. MM cells may also become sensitive to bortezomib in the presence of nelfinavir. Nelfinavir demonstrates inhibitory effects on the proteasome, an activity that bortezomib fails to target. This leads to the hypothesis that nelfinavir’s anti-neoplastic actions in MM cells are mostly mediated by this mechanism in conjunction with suppression of Akt-phosphorylation [76]. An Open-Label Phase I clinical trial is being conducted to investigate the effects of combining nelfinavir and bortezomib in patients with relapsed and refractory MM [77].

### 3.9. Metformin

Metformin is a commonly prescribed medication for the treatment of insulin-independent DM type 2 (diabetes mellitus). AMPK is crucial for cellular metabolism, specifically in controlling glucose metabolism. Metformin enhances the utilization of blood glucose absorption by muscles and liver by activating AMPK. AMPK suppresses mTORC1, a pathway associated with cell proliferation, while activating p53, a tumor suppressor protein involved in promoting cell death. Metformin has been found to inhibit mTORC1 in a way that is different from AMPK [142]. Metformin has been found to have various effects on cancer cells, including inhibiting epithelial-to-mesenchymal transition (EMT), inducing senescence, and reducing the survival of cancer stem cells [143]. In MM, the development of EMT phenotypes is pertinent and has been shown to be the hypoxic drive and contributing to malignant plasma cell migration and cells acquiring drug resistance [144]. MM cells will also overexpress EMT transcription factors in response to signals from the BMM [144]. Therefore, metformin has the potential to be developed as a therapeutic anti-cancer treatment [145,146]. Based on a comprehensive analysis of observational studies, it was found that individuals with diabetes who were prescribed preventative metformin medication experienced a lower likelihood of developing cancer and succumbing to cancer-related causes [145]. Studies suggest that metformin may have inhibitory effects on various cell cycle regulatory proteins, such as cyclin D1, Rb, ERK1/2, JAK2/STAT signaling, and mitochondrial function, which could potentially explain its observed effects [147]. Metformin suppresses IL-6 signaling by reducing IL-6R expression on MM cells [148]. In addition, metformin is known to stimulate autophagy. Researchers are currently focusing on the AMPK/mTORC1 and mTORC2 pathways to trigger autophagy and cell cycle arrest in myeloma. Importantly, in comprehensive studies, metformin use among diabetic individuals with MGUS was linked to a decreased risk of developing active MM [149,150]. Several preclinical studies have also shown the anti-myeloma effects of metformin [79,151]. Several studies have demonstrated the synergistic effects of combining metformin with drugs that affect glycolysis, such as ritonavir in MM [79]. The interaction between metformin and a Glut4 glucose transporter inhibitor, which would modulate MM, was predicted in silico [80]. Metformin and FTY720 synergistically trigger apoptosis in MM cells [152]. Metformin has been found to potentially speed up cell death in MM cells when combined with the proteasome inhibitor bortezomib, by inhibiting a protective autophagic response and disrupting protein homeostasis [81]. A Phase I clinical trial, conducted in an open-label manner, investigated the combination of metformin and bortezomib in patients with relapsed and refractory MM [77]. It appears that when metformin is used alongside chemotherapy, MM patients may experience a longer survival period.

### 3.10. Bisphosphonates

Bisphosphonates are a class of chemical compounds that contain two phosphonate groups and two R-groups, distinguishing them from other bisphosphonates. There are two main classes of bisphosphonates: nitrogenous and non-nitrogenous. Examples of nitrogenous bisphosphonates are clodronate, tiludronate, etidronate, and nitrous bisphosphonates. On the other hand, non-nitrogenous bisphosphonates include zoledronate, neridronate, alendronate, pamidronate, ibandronate, olpadronate, and risedronate. Due to its early presence in bone metabolism research, bisphosphonates were later recommended for postmenopausal women as a preventive measure against bone loss (osteoporosis). In addition, its application was broadened to encompass patients suffering from bone loss caused by metastatic lung cancer, breast cancer, and MM. The mechanisms governing nitrogenous and non-nitrogenous bisphosphonate compounds have distinct differences. The integration of first-generation non-nitrogenous bisphosphonates with the ATP of osteoclasts results in the production of ATP analogs that exhibit resistance to hydrolysis. The activity of farnesyl pyrophosphate synthase (FPPS), a crucial enzyme in the HMG-CoA pathway responsible for cholesterol production, is suppressed by fourth and third-generation nitrogenous bisphosphonates. Based on extensive clinical and laboratory research, it has been found that nitrogenous bisphosphonates are effective in treating both solid and hematological cancers [153]. Osteoclast cells undergo apoptosis when exposed to bisphosphonates [154]. The growth of MM is facilitated by the favorable conditions found in bone microenvironments. The interaction between osteoclasts and MM cells is reciprocal, with each one influencing the other’s activity and survival [82,83]. The collaboration between MM cells and osteoclasts is disrupted by bisphosphonates, which creates an unfavorable microenvironment for MM cell growth. In MM, bisphosphonate has potential to serve as a valuable supplemental treatment.

### 3.11. CuET

Diethyldithiocarbamate-copper complex (CuET) is a product of metabolism combining Disulfiram (DSF) and cofactor copper [155]. DSF is an FDA-approved drug for alcohol abuse, and it causes adverse, immediate, and unpleasant reactions to alcohol consumption [156]. This is because DSF inhibits enzyme aldehyde dehydrogenase (ALDH), which is crucial in the liver’s metabolism of alcohol [156]. CuET is a biologically active complex, due to the copper cofactor, and is shown to produce intracellular and extracellular reactive oxygen species (ROS), which promotes apoptosis [157]. Additionally, in cancer cells, CuET caused levels of proteasome inhibition, which MM cells are extremely sensitive to due to their function in antibody production, and aggregated proteins such as IκB (which inhibits NF-κB), p27, Kip1, and c-Myc [157]. Multiple myeloma stem cells (MMSCs), cells integral in the progression of disease, display increased ALDH activity and have higher tumorigenic rates [158]. Additionally, the dominant isoform of ALDH is notably more expressed in bortezomib-resistant MM cells, and a significantly higher percentage of the resistant cells express ALDH [158]. Inhibition of the metabolic enzyme offers a viable treatment application for MM. An in vitro and in vivo study observed that CuET can negatively affect the stem cell qualities of MM cells, minimize tumor growth, and eliminate clonogenicity by inhibiting ALDH via the Hedgehog and ALDH1A1 pathways [159]. Furthermore, CuET successfully induced apoptosis in bortezomib- and carfilzomib-resistant MM cells and affected MM cells in ways similar to proteasome inhibitors as it induced the unfolded protein response [84]. CuET’s mechanism of action and effects on MM cells in vitro and in vivo establish the drug as a promising target to be repurposed for MM treatment.

### 3.12. Albendazole

Albendazole (ABZ) is an antiparasitic agent used to treat intestinal worm infections and filarial infections [160]. ABZ prevents the uptake of glucose by parasites and mammalian cells through the inhibition of the microtubule systems [160]. However, the drug has applicable anti-cancer properties. In accordance with its mechanism in parasites, in leukemia cells, ABZ downregulates the protein SIRT3 to cause upregulation of TNF-α, which subsequently destabilizes microtubules within the cell, halting the cell cycle [161]. Additionally, ABZ-treated leukemia cells showed evidence of ROS production, activation of the death receptor-mediated pathway, and activation of p38 MAPK, which ultimately resulted in apoptosis of the malignant cells [161]. Additionally, ABZ-treated leukemia cells showed evidence of ROS production. These apoptotic effects are consistent in MM cells both in vitro and in vivo [85]. Among MMSCs, ABZ reduced the number of ALDH-positive cells, which correlates with higher rates of tumorigenesis and acquired resistance, and in doing this, resistant MM cells were resensitized to bortezomib [85]. Furthermore, treatment of MM cells with ABZ inhibited the NF-κB pathway by diminishing transcription factor p65 [85]. Repurposing ABZ to be utilized in relapsed MM treatments offers a promising outlook given the current evidence.

## 4. Conclusions and Future Prospectives

The search for effective treatments for MM continues to be a substantial problem, requiring ongoing investigation into new therapeutic approaches. Drug repurposing, the process of reassessing existing pharmaceuticals for new applications, offers a promising approach that circumvents numerous difficulties in drug development and expedites advancements in anti-cancer treatment. Drug repurposing is essential in the advancement of innovative anti-cancer therapies. This article explores different medications utilized as anti-cancer agents and their prospective applications in the treatment of MM. Although drugs like thalidomide have been successful in clinical settings, others such as statins, aspirin, metformin, and clarithromycin have shown encouraging preclinical and early clinical results (outlined in Table 2). This highlights the importance of conducting additional research and clinical trials to fully harness their potential in enhancing outcomes for patients with MM. Thorough inquiry is necessary to translate non-cancer medicines into effective treatments for MM. Prior to implementing repurposed medications in patient care, it is crucial to conduct thorough clinical trials to determine their effectiveness and mechanism of action. Future research should concentrate on studying the most effective dosage of repurposed medications specifically designed for people with MM. In addition, investigating possible synergies with conventional MM treatments could improve therapy results and inform the creation of improved treatment plans. Hence, it is imperative to have a comprehensive understanding of the actions of medications prior to subjecting cancer patients to testing. Utilizing cancer and drug databases, such as the one described in Table 1, can facilitate, and accelerate this procedure. The fundamental issues regarding the efficacy and safety of drugs in clinical trials have remained unchanged over time. Drug repurposing offers significant financial and time efficiencies, making it an appealing option that can potentially save huge resources when compared to typical drug development pathways. Repurposing medications that have failed in Phase II or III studies for their original intended uses, by utilizing their shown anti-cancer effectiveness, could expand the range of treatment choices for MM. Around 48% of the overall expenses for drug development are allocated to preclinical development and Phase I clinical trials of a new chemical entity (NCE). The duration of these critical phases normally spans an average of seven years [8]. Many drugs fail during the transition from Phase I to Phase II clinical trials because they are either ineffective or harmful. Hence, medications that fail Phase II trials may be seen as possible contenders for repurposing if they exhibit anti-cancer properties. Although non-cancer drugs have shown encouraging effects in preclinical studies, their clinical application is still questionable. To fully harness the benefits of repurposing drugs for MM patients, it is imperative to address these concerns by doing thorough clinical studies. Ultimately, the potential of medication repurposing to further MM treatment is contingent upon thorough validation through rigorous clinical studies. These endeavors are crucial for fully realizing the potential of repurposed medications in enhancing outcomes for MM patients.

## 5. Practice Points

Although there have been numerous clinical trials conducted to evaluate different approaches for treating cancer, the 5-year survival rate for individuals with MM in the US remains at a modest 55%.Myeloma remains a challenging malignancy to treat due to the development of drug resistance, resulting in relapse for all patients.There is an ongoing demand for new medications. However, the process of finding a new treatment can often be quite time-consuming. Therefore, repurposing already approved non-cancer medication for MM can aid in the discovery of new effective drugs.The potential for repurposing approved drugs is promising, although a thorough analysis of these agents is necessary before they can be considered for clinical trials.

## 6. Research Agenda

The potential of various non-anti-cancer drugs as an anti-myeloma treatment was discussed.Thalidomide stands out as an exemplary repurposed agent for treating MM.There is encouraging evidence that statins, rapamycin, clarithromycin, and leflunomide can inhibit MM.Extensive animal studies using the MM animal model, along with phase 1 clinical studies, are necessary to thoroughly investigate these agents as potential MM therapies.

## Figures and Tables

**Figure 1 cancers-16-02381-f001:**
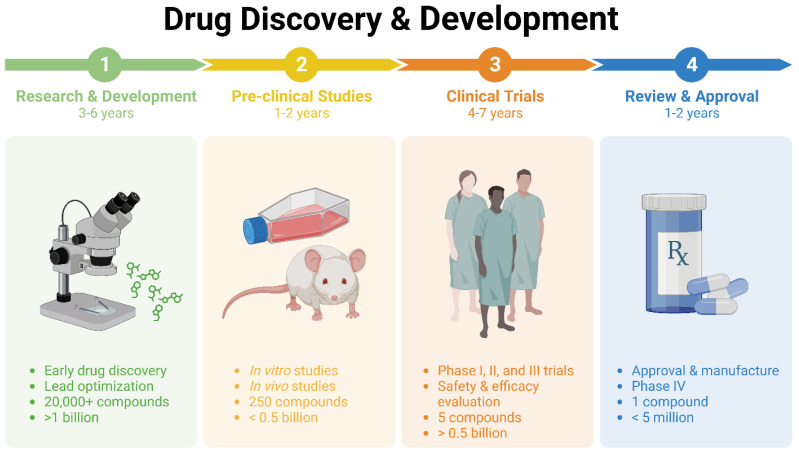
Drug discovery and development stages and process.

**Figure 2 cancers-16-02381-f002:**
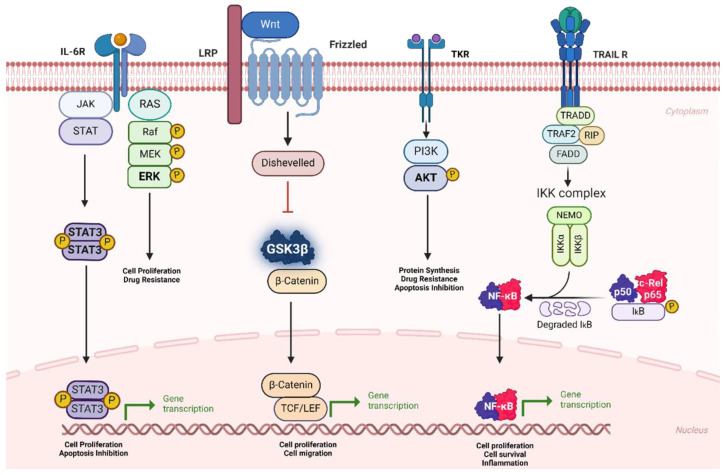
The key signaling pathways involved in the development of MM.

**Figure 3 cancers-16-02381-f003:**
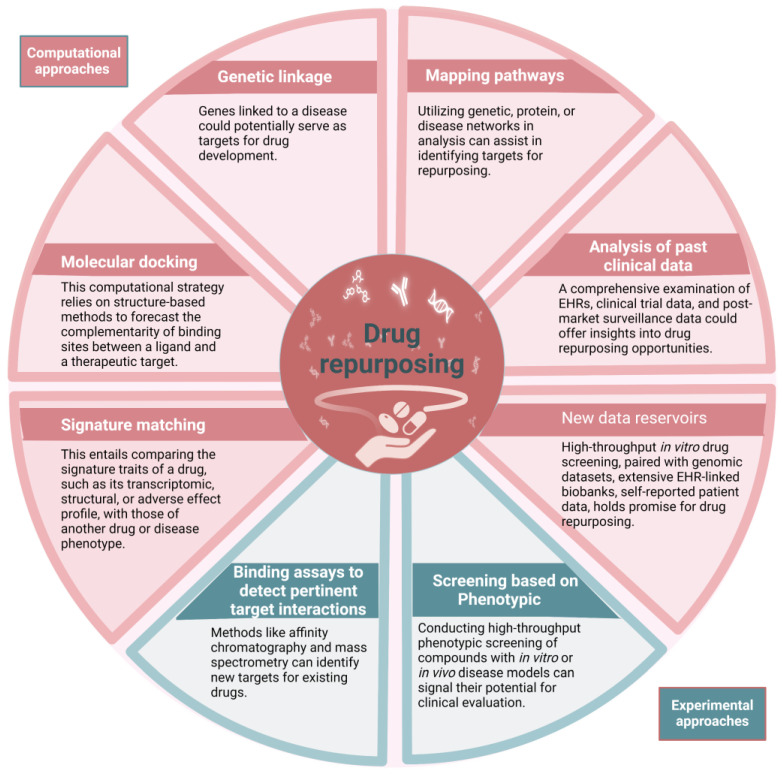
Drug repurposing employs diverse computational approaches, either independently or in concert, to methodically scrutinize various large-scale datasets for insightful interpretations regarding repurposing hypotheses. Additionally, experimental approaches can uncover repurposing prospects. EHR stands for electronic health record.

**Figure 4 cancers-16-02381-f004:**
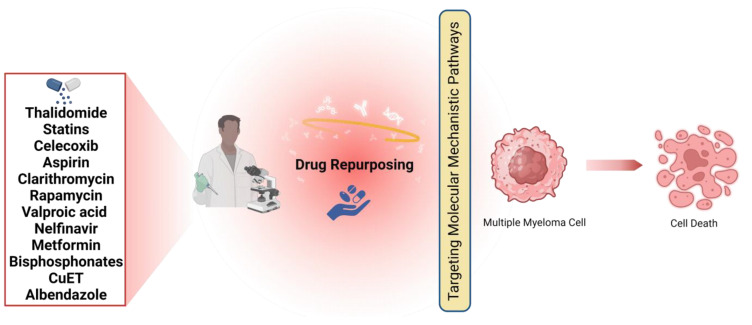
Drug candidates for repurposing to treat multiple myeloma (MM).

**Table 1 cancers-16-02381-t001:** Resources for identifying drug interactions that may possibly be employed in drug repurposing.

Purpose	Resource	Refs.
Human pathways and protein–protein interaction (PPI)	BiGRID, STRING, HAPPI, KEGG, Reactome	[27,28,29,30,31]
Molecular classification of more than 20,000 main cancers matched normal tissue from 33 types of cancer	Cancer Genome Atlas	[32]
Protein expression in cancer, matched normal tissues, and human cancer cell lines	The Human Protein Atlas	[33,34,35]
Drug sensitivity, gene expression, and genotype for human cancer cell lines	Cancer Cell Line Encyclopedia	[36]
Data of genome-wide transcription expression from cultured human cancer cells with many small compounds	Connectivity Map 02 (CMap)	[37,38]
Disease-specific gene curation and analysis	OMIM, GEO	[39,40]
Disease–disease connectivity; connectivity of two genes elaborated within the same disease	The human disease network	[40]
Disease similarities as seen through the lens of gene regulatory mechanisms; comprehension of disease etiology and pathophysiology	Human Disease Network Database (DNetDB)	[41]
Drug–drug interaction; comprehensive drug-target information on tens of thousands of drugs and targets	DrugBank	[42]
Drug–drug interaction	SFINX	[43]
Database of more than 270 non-cancer drugs for potential repurposing for anti-cancer therapy	Repurposing Drugs in Oncology (ReDO)	[44]
Database of drugs and adverse drug reactions (ADRs)	Side Effect Resource (SIDER)	[45]
Withdrawn or discontinued drugs	WITHDRAWN	[46,47]
An inventory of main and secondary uses for repurposed pharmaceuticals	RepurposeDB	[48]
Chemical (including drugs)–protein interaction network	STITCH	[47]
Data on the sensitivity of hundreds of compounds and over a thousand cancer cell lines	Genomics of Drug Sensitivity in Cancer (GDSC)	[49]
Gene expression pattern-based prediction of drug effectiveness against cancer	DeSigN	[50]

**Table 2 cancers-16-02381-t002:** Drugs being repurposed for multiple myeloma (MM).

Drug Name	Old-Indication	New-Indication	Mechanism of Action	Clinical Trials Status	Refs.
Thalidomide	Sedative, anti-nausea	MM	Inhibits IKK (also NF-κB); inhibits TNF; inhibits IL-1, IL-6, IL-12, VEGF	Approved in combination with dexamethasone	[51,52]
Statins	High Cholesterol	MM	HMG-CoA reductase inhibitors, upregulation of PUMA and NOXA	Smouldering MM, phase II	[53,54,55,56]
Celecoxib	Anti- inflammatory	MM and drug-resistant MM	Inhibits COX-2, inhibits Mcl-1, Bcl-2, survivin, Akt	Not for MM, approved for FAP	[57,58,59]
Aspirin	Anti- inflammatory	MM	Inhibits COX-1 and COX-2, suppresses cytokines and NF-κB, inhibits EKR and Blimp1, activates ATF4/CHOP	Preclinical	[60,61]
Artesunate	Malaria	MM and drug-resistant MM	Decreased expression of MYC and Bcl-2, triggers cleavage of caspase-3	Preclinical	[62,63,64,65]
Leflunomide	Rheumatism	MM	DHODH inhibitor, cyclin D2 and pRb inhibition	Phase II	[66,67]
Clarithromycin	Antibiotic	MM and drug-resistant MM	Inhibits IL-6 and MGFs	Phase III	[68,69]
Rapamycin	Fungal infections	MM	Antagonist of mTOR	Phase I	[70,71,72,73]
Valproic acid	Seizures, migraine, and epilepsy	MM	Blocks HDAC, inhibits NF-κB and cytokines	Preclinical	[74]
Nelfinavir	HIV Infection	MM and drug-resistant MM	Inhibits 26S proteasome- disrupts Akt and STAT3, ERK1/2	Phase I	[75,76,77,78]
Metformin	Diabetes mellitus type 2	MM	Activates AMPK (suppresses mTORC1, activates p53), inhibits EMT, regulates cell cycle proteins (ERK1/2, JAK2/STAT), IL-6 suppression	Smoldering Myeloma and Monoclonal gammopathy of undetermined significance phase II, MM phase I	[77,79,80,81]
Bisphosphonates	Osteoporosis	MM	HMG-CoA pathway suppression, osteoclast apoptosis	Preclinical	[82,83]
CuET	Alcohol-abuse drug disulfiram (DSF)	Drug-resistant MM	ALDH inhibition	Preclinical	[84]
Albendazole	Parasitic infections	Drug-resistant MM	Microtubule system interference, p65/NF-κB pathway inhibition	Preclinical	[85]

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
