# Peer review of "Discovering Potential in Non-Cancer Medications: A Promising Breakthrough for Multiple Myeloma Patients"

_cancers, 2024, doi:10.3390/cancers16132381_

Round 1

Reviewer 1 Report

Comments and Suggestions for Authors

The manuscript by Omar S. Al-Odat et al. presents a review of the repurposing of approved non-oncology drugs for the treatment of multiple myeloma as a promising strategy to address the urgent need for effective and affordable cancer treatments.

The authors highlight the importance of using drug repurposing compared to new drug development, which is a multi-step process that involves identifying a therapeutic drug molecule that is clinically effective in treating a disease.

This study discusses various drugs used as anti-cancer agents and potential pharmaceuticals for MM treatment.

Overall the manuscript is well written, but there are some typos and inappropriate terms:

-          Line 198 … cells consistently overexpress the pathways … (a pathway is a series of chemical reactions in which a group of molecules in a cell work together to control the cell's function; some proteins may be activated and others repressed)

-          Line 227 Cox-2 is not pro-inflammatory cytokine. (Cyclooxygenase-2 (COX-2) is Prostaglandin-endoperoxide synthase 2 (HUGO PTGS2), is an enzyme that involved in the conversion of arachidonic acid to prostaglandin H2, an important precursor of prostacyclin, which is expressed in inflammation.)

Some comments on Table 1:

It is a list of web resources.

Could the authors provide examples of which drugs for repurposing in MM were selected using "Human Pathways and Protein-Protein Interaction (PPI)", "Predicting Cancer Drug Efficacy Based on Gene Expression Patterns"? To be more related to the topic of the review, it would be appropriate to describe the application of how they help to select repurposing of drugs for MM therapy.

In title of manuscript A New Hope for Multiple Myeloma: would be good to writ A New Hope for Multiple Myeloma Patients

The manuscript is suitable for publication after the above comments have been corrected.

Comments on the Quality of English Language

The English is good, but there are some typos.

Author Response

The manuscript by Omar S. Al-Odat et al. presents --------- effective and affordable cancer treatments.

The authors highlight the importance of using drug repurposing ---------- MM treatment. Overall, the manuscript is well written, but there are some typos and inappropriate term:

Response: The authors appreciate the reviewer's statement that the manuscript is well written and emphasize the need of adopting drug repurposing rather than new drug development for the treatment of MM. We have modified the manuscript as suggested by reviewer.  

Comment #1: Line 198 … cells consistently overexpress the pathways … (a pathway is a series of chemical reactions in which a group of molecules in a cell work together to control the cell's function; some proteins may be activated and others repressed)

Response:  Authors appreciate your suggestion. We revised the entire paragraph to enhance its appeal and eliminated any instances of “overexpression” that were incorrectly used.

Comment #2: Line 227 Cox-2 is not pro-inflammatory cytokine. (Cyclooxygenase-2 (COX-2) is Prostaglandin-endoperoxide synthase 2 (HUGO PTGS2), is an enzyme that involved in the conversion of arachidonic acid to prostaglandin H2, an important precursor of prostacyclin, which is expressed in inflammation.

Response:  We removed the pro-inflammatory word before COX-2. Thanks.

Comment #3: Could the authors provide examples of which drugs for repurposing in MM were selected using "Human Pathways and Protein-Protein Interaction (PPI)", "Predicting Cancer Drug Efficacy Based on Gene Expression Patterns"? To be more related to the topic of the review, it would be appropriate to describe the application of how they help to select repurposing of drugs for MM therapy.

Response: Authors value and acknowledge this suggestion. We have revised the text and included specific instances of leveraging biodata in the discovery of medications.

Comment #4: In the title of the manuscript, A New Hope for Multiple Myeloma: it would be good to write A New Hope for Multiple Myeloma Patients.

Response: Based on the feedback and comments from other reviewer, we have revised the title to “Discovering Potential in Non-Cancer Medications: A Promising Breakthrough for Multiple Myeloma Patients”.

Reviewer 2 Report

Comments and Suggestions for Authors

The authors report a new hope for multiple myeloma.

1.     The authors should provide in Figure the signaling of multiple myeloma in the first part of the section.

Author Response

Comment #1: The authors should provide in Figure the signaling of multiple myeloma in the first part of the section.

Response: Thank you for the suggestion. We added a new figure as Figure 2 in the modified text.

Reviewer 3 Report

Comments and Suggestions for Authors

The authors list and review potentially effective drugs (approved for other than cancer) for multiple myeloma. These results have the potential to encourage patients and portend the arrival of more new drugs. However, there are currently not enough reports on these drugs for myeloma, and these drugs are expected to be further investigated in the future. This is a useful paper that summarizes the current information about non-cancer drugs for MM.

Author Response

The authors list and review potentially effective drugs (approved for other than cancer) for multiple myeloma. These results have the potential to encourage patients and portend the arrival of more new drugs. However, there are currently not enough reports on these drugs for myeloma, and these drugs are expected to be further investigated in the future. This is a useful paper that summarizes the current information about non-cancer drugs for MM.

Response: The authors are pleased to receive favorable feedback and appreciate the recognition of the significance of our study. Authors value your evaluation of our manuscript as valuable in consolidating up-to-date knowledge on non-cancer medicines for MM.

Reviewer 4 Report

Comments and Suggestions for Authors

The review entitled: “A New Hope for Multiple Myeloma: Harnessing the Power of Non-Cancer Medications(D: cancers-3050539) by Al-Odat et al. highlights a noncomprehensive list of conventional therapies with a potentially significant anti-myeloma properties and anti-neoplastic effects. Additionally, the authors aim to offer valuable insights into the resource for drug repurposing in MM.

Albeit the review is well written and of special interest, comments should be addressed to further improve the manuscript.

Comments:

1.     The authors explore in their review the potential of various medications to treat MM and highlighting the anticancer effects of well established and effective therapies such as thalidomide and bisphosphantes compared to medications e.g. statins, aspirin and clarithromycin which are not quite comparable in term of efficacy in MM therapy. Therefore, the title of the review suggests too much “hope” and should be adapted.

2.     Albeit the aim of the review is comprehensible and important, the overall purpose of the review is not quite clear. Since the development/repurpose of thalidomide was an important cornerstone in the therapy of MM patients, these effects could not be evaluated in medications such as e.g. artesunte or metformin. Therefore, the review should more balanced in terms of the pharmacological repurposing strategies of specific medicines that could repurposed to treat MM patients.

3.     Conclusion: taken into account comment 1 and 2 the conclusions is too hypothetic. Moreover, sentence page 14, line 511-512 is not clear and should be revised.  

4.     Conclusions, practice points. The last point is very important and essential. Therefore, the whole manuscript should be adapted accordingly, since the repurposing process needs additional clinical trials in terms of dose finding, safety and efficacy which as to be taken into considerations.

5.     Conclusion should be more balanced and focus of the unmet clinical need of further treatment option within therapy of MM patients.

6.     Section 2: is too long and should be shortened.

Author Response

The review entitled: “A New Hope for Multiple Myeloma: Harnessing the Power of Non-Cancer Medications” (D: cancers-3050539) by Al-Odat et al. ----- the resource for drug repurposing in MM.

Response: Thank you for your positive assessment and insightful comments on our manuscript.  We have addressed all concerns as described below.

Comment #1: The authors explore in their review the potential of various medications to treat MM and highlighting the anticancer effects of well-established and effective therapies such as thalidomide and bisphosphantes compared to medications e.g. statins, aspirin and clarithromycin which are not quite comparable in term of efficacy in MM therapy. Therefore, the title of the review suggests too much “hope” and should be adapted.

Response: We appreciate your valued comments. After thorough deliberation of your feedback and the recommendation from reviewer 1, we have revised the title to "Discovering Potential in Non-Cancer Medications: A Promising Breakthrough for Multiple Myeloma Patients”. We hope that the new title adequately supports the content of the paper.

Comment #2: Albeit the aim of the review is comprehensible and important, the overall purpose of the review is not quite clear. Since the development/repurpose of thalidomide was an important cornerstone in the therapy of MM patients, these effects could not be evaluated in medications such as e.g. artesunte or metformin. Therefore, the review should more balanced in terms of the pharmacological repurposing strategies of specific medicines that could repurposed to treat MM patients.

Response: We appreciate your perceptive input. We have responded to your comments by modifying the conclusion to present a more equitable viewpoint and integrating more details where needed to improve comprehensibility. In addition, we have augmented the text with additional details and references to certain pharmaceuticals, highlighting their present status in clinical trials and their potential for being repurposed as therapies for MM.

Comment #3: Conclusion: taken into account comment 1 and 2 the conclusions is too hypothetic. Moreover, sentence page 14, line 511-512 is not clear and should be revised. 

Response: We appreciate your feedback regarding the conclusion. In response to your criticism, we rewrote the conclusion section. 

Comment #4: Conclusions, practice points. The last point is very important and essential. Therefore, the whole manuscript should be adapted accordingly, since the repurposing process needs additional clinical trials in terms of dose finding, safety and efficacy which as to be taken into considerations.

Response:  We have revised the conclusion section and made modifications to other sections in order to incorporate additional information whenever feasible. Hope it is more balanced now.   

Comment #5: Conclusion should be more balanced and focus of the unmet clinical need of further treatment option within therapy of MM patients.

Response: We appreciate your feedback on the conclusion part of the manuscript. We have revised the entire section to achieve a more equitable discourse, emphasizing the unfulfilled medical requirements and the significance of additional therapeutic alternatives for patients with MM.

Comment #6: Section 2: is too long and should be shortened.

Response: We appreciate your input regarding Section 2 of our paper. We have thoroughly assessed section 2. We concur that the section is excessively lengthy, and as a result, we have made modifications to it.

Round 2

Reviewer 2 Report

Comments and Suggestions for Authors

none

Reviewer 4 Report

Comments and Suggestions for Authors

The review entitled: “A New Hope for Multiple Myeloma: Harnessing the Power of Non-Cancer Medications(D: cancers-3050539) by Al-Odat et al. highlights a noncomprehensive list of conventional therapies with a potentially significant anti-myeloma properties and anti-neoplastic effects. Additionally, the authors aim to offer valuable insights into the resource for drug repurposing in MM. 

After revisions of the manuscript, the authors addressed all my initial comments sufficiently.